# COVID-19 vaccine uptake and hesitancy survey in Northern Ireland and Republic of Ireland: Applying the theory of planned behaviour

**Gavin Breslin**[1]⊕*, **Martin Dempster**[2]⊕, **Emma Berry**[2]⊕, **Matthew Cavanagh**[1]⊕, **Nicola C. Armstrong**[3]⊕¶

**1** School of Psychology, Ulster University, Coleraine, Northern Ireland, **2** School of Psychology, The Queen's University, Belfast, Northern Ireland, **3** Health and Social Care Research & Development (HSC R&D) Division, Public Health Agency, Belfast, Northern Ireland

⊕ These authors contributed equally to this work.
¶ This work was part of the Public Health Agency, Northern Ireland COVID-19 Behaviour Change Group
* g.breslin1@ulster.ac.uk

**Data Availability Statement:** Data has been made available and will now be hosted here:https://pure.

## Abstract

The Coronavirus Disease (COVID-19) caused by severe acute respiratory syndrome coronavirus 2 (SARS-CoV-2) first appeared in Wuhan, China in late 2019 and since then has caused unprecedented economic and social disruption as well as presenting a major challenge to public health. Despite mass progress in COVID-19 vaccination uptake, vaccine hesitancy or anti-vax information has been reported that can delay public acceptance of a vaccine. An online cross-sectional survey (n = 439) assessed COVID-19 vaccine uptake and hesitancy in adults in Northern Ireland and the Republic of Ireland. Participants completed an adapted version of the Theory of Planned Behaviour Vaccine Questionnaire, the Vaccine Attitudes Scale (VAX), Vaccine Confidence Scale, and questions on previous experience of COVID-19. Results showed that 66.7% of the sample intended to get a vaccination as soon as possible, 27.15% reported they will get a vaccine when others get theirs and when it is clear there are no side effects. 6.15% had no intention of getting a vaccine. Overall, there is a high mean intention (M = 6.12) and confidence to get a COVID-19 vaccine. There was low vaccine hesitancy (M = 2.49) as measured by the VAX scale. A further analysis of the sub factors of the VAX showed there is uncertainty and mistrust of side effects for children. The finding demonstrate that the Theory of Planned Behaviour can be useful in making recommendations for public health considerations when encouraging vaccine uptake and reducing vaccine hesitancy.

## Introduction

The Coronavirus Disease (COVID-19) caused by severe acute respiratory syndrome coronavirus 2 (SARS-CoV-2) first appeared in Wuhan, China in late 2019 (World Health Organisation,

ulster.ac.uk/en/datasets/data-set-for-covid-19-vaccine-uptake-and-hesitancy-survey-in-nort.

**Funding:** The authors received no specific funding for this work.

**Competing interests:** The authors have declared that no competing interests exist

2020) and since then has caused unprecedented economic and social disruption as well as presenting a major challenge to public health (World Health Organisation, 2020). As of March 12th, 2021, the disease has infected more than 118,707,983 people with 2, 631,385 deaths worldwide. In Europe there has been 38,947,362 confirmed cases and 881,973 reported deaths [1]. In the United Kingdom, 4.26 million (England = 3.73m; Scotland = 210K; Wales = 207K; Northern Ireland = 115K) cases of coronavirus and 126,000 (England = 111K; Scotland = 7,510; Wales = 5,454; Northern Ireland = 2099) deaths are reported [1], while in the Republic of Ireland 227K confirmed cases and 4,534 deaths were reported.

To restrict the spread of the virus, and reduce demand on health services, governments worldwide introduced lockdown and quarantine measures, social distancing, and restrictions in face to face education, workplace and commercially available shopping services. The impact of these restrictions has been an increase in unemployment rates, employees being furloughed, business disruption and school and university closure, with children being home schooled by parents/family, and university teaching being delivered online.

## SARS-Cov2 vaccine

The development of an effective SARS-Cov2 vaccine to avoid further human and social, and economic loss was required. Vaccinations are an important method of public health disease prevention involving the administration of a microorganism in a live, killed or weakened state to stimulate immunity against disease (Centers for Disease Control and Prevention, 2018). The development of a vaccine was expedited by the United Kingdom (UK) Government Vaccine Taskforce (VTF) with several SARS-Cov2 vaccine trials undertaken to identify which vaccines are both safe and effective, so that vaccination programmes can start as early as possible.

On the 8th December 2020 the first person in the UK received a Coronavirus vaccination, since then efforts to roll out the vaccine have been encouraging and at pace. As of March 15th, 2021, 24, 453,221 individuals in the UK have received their first vaccine dose, while 1,610,280 have received a second dose. In Northern Ireland 629,461 had received a first dose and 54,636 had received a second dose of a vaccine by 15th March 2021 [2] Vaccine distribution is different between NI and Republic of Ireland. The first dose of the Coronavirus vaccination in the Republic of Ireland was administered on the 29th December 2020 and, as of the 13th March 2021, 451,589 individuals received a first vaccine dose while 164,345 had received a second dose in the Republic of Ireland [3].

Encouragingly, 79% of 140,000 people surveyed in 140 countries indicated vaccines are safe and 73% reported that they trusted a doctor or nurse more than any other source of health advice [4]. From the same survey of UK respondents (n = 1000) 75% felt that vaccines were safe, and 95% of those with children have had their children vaccinated. These figures are similar to those in the Republic of Ireland, where 73% of participants felt vaccines were safe, and 93% have had their children vaccinated. However, safety updates have been issued for both the AstraZeneca [5] and Janssen [6] vaccines that may impact public confidence in COVID-19 vaccine uptake.

## Vaccine hesitancy

Despite the progress in the public health distribution of the vaccinations, there are individuals who will perceive vaccination as unsafe and/or unnecessary [7]. Vaccine hesitancy is defined as the delay in acceptance or refusal of a vaccine despite their availability to the public [8]. Predicting vaccine hesitancy is complex because it is context and time specific, and can vary across vaccines. Pre COVID-19, vaccine hesitancy was listed as one of the top ten threats to global

public health, as a result it was recommended that countries incorporate plans to measure and address vaccine hesitancy into their public health programmes [9].

There are several explanations for vaccine hesitancy, some of which are medical and ethical concerns that have been in existence since the emergence of vaccines in the 1700's [10,11] Recently, in 2018 [12] the WHO identified confidence (i.e. trust in the vaccine safety and service providers), complacency (i.e. lack of perceived benefits of the vaccine and low sense of risk), and inconvenience in accessing vaccines as some of the key reasons underlying vaccine hesitancy. Several psychological factors are demonstrated to influence COVID-19 vaccine hesitancy. These include: low altruistic beliefs about the need to protect others; personality traits such as neuroticism and conscientiousness, conspiracy, religious beliefs, paranoid beliefs, mistrust of authority and the attitudes and behaviour of others (family, friends and health professionals) towards vaccines [13,14].

Factors found to increase hesitancy include: forgetting to register for a vaccine, location of the vaccine centre, misinformation, lack of disabled access, previously declining a vaccine, a preference for natural immunity and worries about unforeseen future side effects of receiving a vaccine.

## Vaccine hesitancy and children

Vaccine hesitancy does not appear to be the same across the population. Over a decade ago hesitancy was reported to be on the rise amongst parents (Gowda & Dempsey, 2013) with concerns and a distrust about the potential side effects vaccines can have on children in the immediate and short-term. This distrust in vaccines for children may be in part due to a discredited case series in the Lancet [15] that suggested measles, mumps and rubella (MMR) vaccine predisposed children to behavioural regression and pervasive developmental disorder [16]. The Lancet publication led to a plethora of studies, refuting the link between autism and vaccination [17,18], although damage to parental confidence, and public opinion on vaccine certainty for children may have already occurred, evident in a measles outbreaks in the UK, USA and Canada in 2008/2009 due in part to parental hesitancy and children not being vaccinated [19]. This finding suggests that public health authorities and effective communication to the public plays an important role in ensuring scientific guidance and information on vaccines is not misleading, and that anti-immunisation rhetoric or fashionable conspiracy theories are downplayed. According to [20] online anti-immunisation videos have gained a large viewing by the public, and social media support. The effects of anti-vaccine media have been shown to influence public attitudes, beliefs, and perceived social norms as predictors of vaccine uptake [21]. A study of teenagers has shown that little is known about how vaccines work, and how booster vaccines for the likes of Meningitis C can protect from the disease. In the same study the participants experiences of immunisation in school were not always positive, however they were enthusiastic at the prospect of introducing more vaccines for their age group [22]. Furthermore, in a qualitative study of teenage girls, parents and health professionals views of the Measles, Mumps, Rubella (MMR), the Human Papilloma virus (HPV) and the Influenza A (H1N1) vaccine showed that having a little bit of doubt about a vaccine, can lead to uncertainties to vaccinations in general, this finding may have implications for hesitancy to receiving the COVID-19 vaccine [23].

## Vaccine hesitancy and COVID-19

Unfortunately, COVID-19 vaccine hesitancy research is limited in both Northern Ireland and the Republic of Ireland making public health decisions regarding communicating effectively with the public more difficult. A study by Murphy et al [13] examined the factors that influence

vaccine hesitancy and uptake. They showed 35% of those in Republic of Ireland and 48.9% in Northern Ireland were hesitant. These figures are particularly concerning given recent research into the contagiousness and rapid spread of the virus, the findings of which suggest that between 55 and 82% of a population should be immune (either via exposure or vaccination) to the virus in order to prevent its spread [24].

The survey by Murphy and colleagues [13] was conducted during the first COVID-19 national lockdown when a vaccine had not been developed, hence views of the public on a future potential vaccine could be very different from views on a vaccine that is available. Furthermore, a small sample size from Northern Ireland (n = 46) took part. To date, no studies have included a psychological behaviour change theory to predict COVID-19 vaccine uptake that would provide a further level of detail when advising national public health authorities. Psychologically oriented informative campaigns have proven effective in promoting health behaviours [25], and a recent rapid review also supports this view [26], therefore the findings of the current study may be beneficial to future campaigns which aim to increase vaccine uptake.

## Psychological theories of behaviour change

Several psychological behaviour change theories have their origins in social, and cognitive sciences, and explain, how and why individuals engage in intentional health behaviours [27,28]. By integrating psychological behaviour change theory such as the Theory of Planned Behaviour (TPB) into survey design of health intention and behaviours, the psychological mechanisms of behaviour change can be better understood, then operationalized when making recommendations on public health messaging (National Institute for Health & Care Excellence, 2018).

The TPB [29] states that an individual's attitudes/beliefs, subjective norms and perceived behavioural control predict intentions and subsequent behaviours. The TPB has been used previously to explain vaccine uptake, although the current study is the first where TPB will be applied to COVID-19 vaccine uptake and hesitancy. As already described, there are many factors that can predict hesitancy and uptake, so in addition to TPB factors, other factors will be included as predictors. These other factors include: *Participant Demographics* (Age, Gender, Employment, Educational Level, Ethnicity), and *Previous Experience of COVID-19* (i.e., having had a positive test for COVID-19, having had to self-isolate, knowing someone who has had COVID-19, knowing someone who has had a vaccine or being at an increased risk of COVID-19). Finally, as mistrust and confidence in the effectiveness of vaccines has been a reported issue for parents consenting to children receiving vaccines, parental mistrust and confidence in children being vaccinated was also included.

## Aims of the study

As there are gaps in the literature in the rates of vaccine uptake and hesitancy, and how to promote vaccine uptake using a psychological theory of behaviour change the study had three aims: 1) To assess SARS-CoV-2 vaccine uptake and hesitancy rates in Northern Ireland and the Republic of Ireland where the healthcare systems are different, yet there is overlap in culture; 2) To assess Attitudes, Subjective Norms and Perceived Behavioural Control as predictors of intentions to vaccinate against COVID-19 and 3) To consider demographic factors, confidence in getting a vaccine and previous experiences of COVID-19 on intention to vaccinate.

## Materials and methods

### Participants

A total of 439 participants took part with 386 (Mean Age = 42.23; SD = 12.16; Range = 19–81; 83% = female, 17% = male) completing all questions.

### Research design

A cross-sectional survey was conducted online via Qualtrics. Participants were recruited via social media platforms Twitter, Facebook and by the Public Health Agency for Northern Ireland's COVID-19 Behaviour Change Cell. No financial incentives were provided to social media organisations or participants for taking part.

### Recruitment

Participants were recruited via social media platforms, Twitter and Facebook. Data was collected via a Qualtrics survey between 29/01/2021–23/02/2021 (i.e. seven weeks after the first COVID-19 Vaccination, and during a national lockdown in NI and ROI).

### Measures

**Previous experience of COVID-19.** All participants were asked whether they: had had a positive test for COVID-19; are at an increased risk of COVID-19; have had to self-isolate; knew someone with COVID-19; knew someone who had received a COVID-19 vaccination.

**Vaccine Confidence Scale [30].** Consists of eight items assessing three factors: benefits of vaccination (Benefits), the harms of vaccination (Harms), and trust in health care providers (Trust). Each item used an 11-point response scale ranging from 0 (strongly disagree) to 10 (strongly agree). The scale is valid and reliable across many diverse populations [30].

**Vaccine Attitudes Examination Scale (VAX) [31].** Consists of 12 items assessing four factors (vaccine mistrust, future worries, profiteering, and preference for natural immunity). Items were presented in the form of statements, with responses on a 6-point Likert-type scale ranging from "strongly agree" to "strongly disagree." Higher scores reflect stronger anti-vaccination attitudes.

**Adapted version of theory of planned behaviour vaccine questionnaire [32].** Consists of 19 items that assess Attitudes (5 items), Subjective Norm (6 items), Perceived Behavioural Control (5 items) and Intentions (3 items) to receive a COVID-19 vaccination. The score on intentions was the main outcome variable to assess behavioural intentions to get the COVID-19 vaccine. Items were presented in a 7-point Likert Scale ranging from "strongly disagree" to "strongly agree". In the absence of a COVID-19 questionnaire, the original scale was adapted to include the word COVID-19 in items when referring to vaccination to make the scale specific to assessment of COVID-19.

An example question assessing Attitudes was 'Using the scales provided, please select the response which reflects your feelings towards the following statements—Getting a COVID-19 vaccine would have definite benefits for my health. For Subjective Norms: 'Using the scales provided, please select the response which reflects your feelings towards the following statement—Most people who are important to me think that I should get a COVID-19 vaccine'. For Perceived Behavioural Control: 'Using the scales provided, please select the response which reflects your feelings towards the following statement—I am confident that if I want to, I will be able to get a COVID-19 vaccine'. Cronbach's alpha reliability coefficient was calculated for TPB variables, Attitudes = .9, Subjective Norm, = .89 Perceived Behavioural Control = .06 and Intention = .94. As Cronbach's alpha level was low for PBC indicating that the items

were weakly related, the scale was recalculated using two items, which showed a moderate correlation (r = 0.4).

## Ethical approval

Ethical approval was granted by Ulster Universities School of Psychology Research Ethics Filter Committee. All participants were provided with a description of the aims and rationale for the study and were asked to provide written informed consent before completing the survey. Participants were free to withdraw at any time. No personal identifying data was collected to ensure confidentiality.

## Data handling

The mean or sum of participants' responses were calculated as per the scoring criteria for each measure. Analysis was conducted using Statistical Package for the Social Sciences 26 (copyright IBM corp., NY, USA) with the alpha level set to $p < .05$. Pearson's correlations were considered weak, moderate and strong when r = .1, .3 and .5 respectively. Effect sizes for t-tests were interpreted using Cohen's d where small, medium and large effects were classed as .20, .50 and .80 respectively. Given the sample size was n = 386, central limit theorem inferred the data was normally distributed. Levene's tests confirmed homogeneity of variances for all statistical tests henceforth. Pearson's bivariate correlations were conducted to assess whether relationships existed between TPB factors, VAX, VCS. Independent samples t-tests assessed whether there was a significant difference in gender or country (NI and ROI). Linear Regression analysis was calculated to establish a best fit model for predicting vaccine intentions.

# Results

## Participant demographics

Demographic information is described in Table 1.

## Experience of COVID-19

When participants were asked about COVID-19, 10.6% reported having had a positive test for COVID-19, 43.3% reported that they had to self-isolate. Almost 20% of the sample considered themselves to be in an increased risk category for Covid-19, 56% said a family member or friend had had COVID-19, and 87.6% reported that they knew someone who had received the COVID-19 vaccine.

## Intentions, attitudes and confidence levels towards receiving a COVID-19 vaccine

Almost 67% intended to get a vaccine as soon as possible, 27.2% reported that they will get their vaccine when others receive theirs and when it is clear there are no side effects, and 6.2% had no intention of getting a vaccine. Descriptive data for the VCS, VAX and TPB factors is presented in Table 2.

There was a high level of confidence (7.98) in getting the COVID-19 vaccine as indicated in the CVS scale. There was also a high mean intention score (6.12) to get a COVID-19 vaccine. A low mean score (2.49) on the VAX scale indicated low vaccine hesitancy in the sample. The mean scores for the sub-factors of the VAX scale were calculated. Presented in order from highest to lowest, these included: worry about unforeseen future events as a result of the vaccine (mean = 3.32, SD = .82), preference for natural immunity (mean = 2.44, SD = .9),

**Table 1. Sample demographics.**

| Participants (n = 386) | | % |
|---|---|---|
| Gender: | Male | 17 |
| | Female | 83 |
| Country: | Northern Ireland | 53 |
| | Republic of Ireland | 43 |
| | Other (e.g. England, USA, Germany) | 5 |
| Ethnicity: | White | 99 |
| | Hispanic Latino | .3 |
| | Black | .3 |
| | Mixed Race | .3 |
| Educational Achievement: | Bachelor's Degree | 7.3 |
| | Master's Degree | 13.2 |
| | PhD or Higher | 11.7 |
| | Trade/School | 39.1 |
| | Prefer not to say | 28.8 |
| Employment Status: | Employed full-time | 53.4 |
| | Employed part-time | 19.2 |
| | Retired | 9.6 |
| | Unemployed seeking employment | 5.3 |
| | Furloughed as a result of COVID-19 | 6.7 |
| | Prefer not to say | 6 |

pharmaceutical company profiteering (mean = 2.17, SD = .98) and mistrust of vaccine benefit (mean = 1.97, SD = .89).

The mean scores for the TPB sub factors (Perceived Behavioural Control, Subjective Norms and Attitudes) indicated that attitudes, PBC and subjective norms contributed to deciding to get a vaccine. No gender effects were found for any of the scales (p>.05), however there was a significant positive correlation between age and intention to get the vaccine (r = .2, df = 331, p < .01).

## Relationship between CVS, VAX and TPB scales on intention to get vaccinated

Pearson Product Moment bivariate correlations were calculated for each scale (VCS, VAX, TPB (perceived behavioural control, subjective norm, attitudes towards COVID-19 vaccine) with intention to vaccinate against COVID-19 (see Table 3).

Confidence in the COVID-19 vaccine was positively correlated with intention to receive the vaccine. Attitudes, perceived behavioural control and subjective norms were positively

**Table 2. Mean score, standard deviation, range and possible range for the Vaccine Confidence Scale, Vaccine Attitudes Examination (VAX) scale and theory of planned behaviour subscales (Attitudes, subjective norms, perceived behavioural control and intention to get a vaccine) are presented.**

| | M | SD | Range | Possible Range |
|---|---|---|---|---|
| Vaccine Confidence Scale | 7.98 | 1.79 | 1–10 | 1–10 |
| Vaccine Attitudes Examination (VAX) Scale | 2.49 | .77 | 1–5 | 1–5 |
| TPB Attitudes scale | 5.66 | 1.38 | 1–7 | 1–7 |
| TPB Subjective Norm scale | 5.64 | 1.34 | 1–7 | 1–7 |
| TPB Perceived behavioural control | 5.66 | 1.38 | 1–6 | 1–7 |
| TPB Intention to get a vaccine | 6.12 | 1.5 | 1–7 | 1–7 |

**Table 3. Pearson product moment bivariate correlations between VCS, VAX and TPB scales.**

| | | VCS | VAX | TPB Perceived Behavioural Control | TPB Subjective Norms | TPB Attitudes | TPB Intention to get a vaccine |
|---|---|---|---|---|---|---|---|
| VCS | Pearson Correlation | 1 | -.809** | .5** | .629** | .757** | .7** |
| | Sig. (2-tailed) | | < .001 | < .001 | < .001 | < .001 | < .001 |
| | N | 361 | 349 | 333 | 333 | 333 | 333 |
| VAX | Pearson Correlation | -.809** | 1 | -.568** | -.622** | -.769** | -.720** |
| | Sig. (2-tailed) | < .001 | | < .001 | < .001 | < .001 | < .001 |
| | N | 349 | 350 | 334 | 334 | 334 | 334 |
| TPB Perceived Behavioural Control | Pearson Correlation | .219** | -.223** | 1 | .285** | .301** | .277** |
| | Sig. (2-tailed) | < .001 | < .001 | | < .001 | < .001 | < .001 |
| | N | 333 | 334 | 334 | 334 | 334 | 334 |
| TPB Subjective Norms | Pearson Correlation | .629** | -.622** | .539** | 1 | .732** | .669** |
| | Sig. (2-tailed) | < .001 | < .001 | < .001 | | < .001 | < .001 |
| | N | 333 | 334 | 334 | 334 | 334 | 334 |
| TPB Attitudes | Pearson Correlation | .757** | -.769** | .574** | .732** | 1 | .827** |
| | Sig. (2-tailed) | < .001 | < .001 | < .001 | < .001 | | < .001 |
| | N | 333 | 334 | 334 | 334 | 334 | 334 |
| TPB Intention to get a vaccine | Pearson Correlation | .700** | -.720** | .561** | .669** | .827** | 1 |
| | Sig. (2-tailed) | < .001 | < .001 | < .001 | < .001 | < .001 | |
| | N | 333 | 334 | 334 | 334 | 334 | 334 |

**. Correlation is significant at the .001 level (2-tailed).

correlated with intention to get the vaccine. Anti-vaccination attitudes as measured by the VAX were negatively correlated with intention to get a vaccine.

To identify what VAX items are correlated with intention to vaccinate, the four subfactors of the VAX scale i) mistrust of vaccine benefit, ii) worries about unforeseen future events, iii) concerns about future profiteering, and iv) preference for natural immunity were correlated with intention to get a vaccine (See Table 4). All four factors negatively correlated with intention to vaccinate, with mistrust of vaccine benefit showing the largest correlated coefficient.

To determine what factors predicted intention to vaccinate, a Linear Regression model was calculated using questions on gender and experiences of COVID-19, CFS, VAX and TPB variables (See Table 5). The regression model explained 71% of the variance in intentions (F

**Table 4. Pearson product moment correlations between VAX subscales and intention to get a vaccine.**

| | | VAX mistrust factor | VAX worry factor | VAX profit factor | VAX immunity factor | Intentions Mean for all questions |
|---|---|---|---|---|---|---|
| TPB Intention to get a vaccine | Pearson Correlation | -.706** | -.521** | -.631** | -.536** | 1 |
| | Sig. (2-tailed) | < .001 | < .001 | < .001 | < .001 | |
| | N | 334 | 334 | 334 | 334 | 334 |

**. Correlation is significant at the 0.01 level (2-tailed).

**Table 5. Regression model.**

| | Unstandardized Coefficients | | Standardized Coefficients | t | Sig. |
|---|---|---|---|---|---|
| | B | Std. Error | Beta | | |
| Constant | 5.680 | .905 | | 6.274 | < .001 |
| Gender | .040 | .121 | .010 | .334 | .739 |
| Have you had a positive test for COVID-19? | .187 | .162 | .039 | 1.161 | .247 |
| Have you had to self-isolate at any point during the COVID-19 pandemic? | .108 | .101 | .035 | 1.069 | .286 |
| Are you in an increased risk category for COVID-19? (e.g. ongoing illness) | -.094 | .113 | -.025 | -.830 | .407 |
| Has anyone close to you (i.e. family member or friend) had COVID-19? | .017 | .095 | .006 | .179 | .858 |
| Do you know anyone who has had a vaccine for COVID-19? | .196 | .149 | .042 | 1.315 | .189 |
| PBC | .104 | .043 | .095 | 2.405 | .017 |
| Sub Norms | .092 | .052 | .082 | 1.766 | .078 |
| Attitudes | .570 | .064 | .523 | 8.847 | < .001 |
| VAX mistrust factor | -.194 | .090 | -.113 | -2.155 | .032 |
| VAX worry factor | .031 | .076 | .017 | .401 | .688 |
| VAX profit factor | -.110 | .080 | -.070 | -1.370 | .172 |
| VAX immunity factor | -.019 | .075 | -.011 | -.250 | .803 |
| VCS | .069 | .047 | .080 | 1.476 | .141 |

a. Dependent Variable: Intentions.

(14,317) = 58.79, p < .001). Attitudes, perceived behavioural control and the VAX mistrust factors were the statistically significant variables explaining COVID-19 vaccine intention. High VAX mistrust scores suggested a lower intention to get vaccinated against COVID-19, whilst higher perceived behavioural control and attitude scores suggested a higher intention to get vaccinated against COVID-19.

## Discussion

The current study showed that attitudes towards COVID-19 vaccines, perceived behavioural control and mistrust predict the high mean intention to get the vaccine in this sample. These findings support the application of the Theory of Planned Behaviour to explain intentions to vaccinate against COVID-19. The data also showed that in the sample there was a high level of confidence to get a COVID-19 vaccine (VCS) and a low vaccine hesitancy score (M = 2.49). This resonates with the confidence factor identified by the WHO as a key predictor of vaccine hesitancy and corroborates the findings of Murphy et al. [14] that psychological factors should be monitored when predicting vaccine uptake, and extends the rationale for the inclusion of the Theory of Planned Behaviour as a suitable basis on which to interpret vaccine uptake and hesitancy.

The finding that attitudes and perceived behavioural control is in keeping with previous research findings which showed that having positive beliefs towards vaccines, feeling in control of getting a vaccine and having people in your close circle of family or friends, especially those who you respect, who received a vaccine influences vaccine acceptance [33]. The current study extends previous studies on vaccine hesitancy to COVID-19, that was previously lacking. The perception that friends and family are pro-vaccination can influence attitudes and vaccine uptake, and the belief that others wanted them to receive a vaccination led them to be more likely to vaccinate [34].

Mistrust in vaccines has also been noted in a recent study conducted in Italy, were willingness to vaccinate against COVID-19 is correlated with trust in the research conducted and in

vaccine effectiveness [35]. This study highlighted the decrease in Italian citizen's trust in science and vaccination at different stages of the pandemic and noted that the proportion of citizens who appeared to be intentioned to get the COVID-19 vaccination would likely be too small to effectively prevent the spread of the disease, if the 55 to 82% immunity statistic previously cited is to be applied [24].

Much of the attribution of responsibility for the decrease in trust of scientific research and vaccination effectiveness is linked to both national and international media, whose coverage of the debate surrounding vaccination will frequently lead to misunderstanding and mistrust of vaccination if not adequately accompanied by appropriate health education [36]. This could be related to the finding that a lack of confidence may predict vaccine hesitancy. Appropriate health education in this context also extends to the description of the reasoning behind certain preventative measures such as school and work closures.

Since the current survey was conducted, recent Astra Zenaca blood clotting reporting in the UK media, and the temporary withdrawal of the vaccine in Republic of Ireland may undermine confidence in vaccine uptake and hence mistrust maybe higher than reported here, highlighting the importance of maintaining positive public views towards vaccine uptake. This public confidence in the Astra Zeneca vaccine may be further damaged by recent recommendations from the Irish National Immunisation Advisory Committee (NIAC) that the vaccine should only be given to those aged over sixty years old [37]. In the absence of psychological informed recommendations, the current study fills this gap.

There are several recommendations from the current study, for example specific targeting of the hesitant population to change attitudes may best be achieved through influencing attitudes and mistrust, in practice this could be the generation of clear optics around health messages and campaigns that allow people who are hesitant to see and hear about others receiving the vaccine. Research by Quinn and colleagues [38] has pointed towards significantly higher rates of flu vaccination among American adults of differing race and religion, suggesting that the influence of peer role models and attitudes may transcend ethnicity. Messaging could also focus on reducing worry of the side effects of the vaccine, through the use of coherent and easily understandable information which is concise enough to retain the interest of the lay person whilst effectively conveying the benefits and safety of vaccinations (e.g. posters, advertisements etc.). We also highlight that the mutation of the SARS-CoV-2 is uncertain and vaccine effectiveness may depend on continued vaccine developments, therefore continued creative attempts to motivate the population to vaccination maybe required.

As previously cited, a significant majority of the population trust their healthcare providers (HCPs) (The Wellcome Trust, 2019 [4]), meaning any future influencing of attitudes and perceived competence should include some actions aimed towards health professionals and general practitioners (GPs) who are viewed by their patients as models for their own health behaviours [39]. A health promotion campaign spearheaded by such individuals who champion the benefits and positive outcomes of vaccinations should result in an increased uptake and reduced vaccine hesitancy throughout the public, as the trust held by patients in their HCPs begins to outweigh any mistrust or misinformation of vaccination.

An informative platform on the immunisation process may also be of great benefit to both the public and HCPs, and suggestions have already been made for the development of a platform similar to the Swiss INFOVAC academic network [40,41]. Such a platform would be a huge outlet for questions on the immunisation process as well as general queries regarding vaccinations, such as previous, non-COVID-19 rollouts and details of existing programmes.

Vaccine misinformation, which is any false information not backed by evidence, is believed to be common on social media [42,43]. Some form of regulation of media debate regarding vaccination should be considered, as the sensationalism of such a serious topic is the root of

much of the existing vaccine mistrust [36]. This regulation should not restrict people from expressing concerns over vaccine rollouts or preventative measures, as collaboration between the public and science is crucial, especially in the current COVID-19 era [44]. Rather, engaging messages and information should be made available to viewers and social media users who feel the need to educate themselves on any COVID-19 vaccination related issues. Similar information, about COVID-19 in general is already employed by popular social media outlets such as Instagram, although a need for more direct, vaccination-oriented information is needed. A small minority of the vaccine-critical misinformation online is thought to come directly from bots [43] which may also be a useful tool for healthcare professionals to use in order to counteract such misinformation.

The study had several limitations that should be considered when interpreting the findings. The TPB provided a lens to explore the complex psychological and social factors underpinning COVID-19 vaccine hesitancy, however a limit of the TPB is that there is no measure of emotion (e.g., disgust) that have been shown to influence vaccine attitudes [45]. Furthermore, the sample was mainly female (83%) and could have included more males. As the survey was completed 29th January- 23rd February, 7 weeks after the first vaccine was administered in Northern Ireland and 4 weeks after the first vaccine was administered in the Republic of Ireland, it is possible views of the public can change in response to vaccines. Finally, the survey was cross sectional, therefore it is not possible to make causal conclusions between the factors assessed and intention to vaccinate. Therefore, further follow-up with longitudinal designs are required and with measures that go beyond assessing intentions, but also incorporate behaviour.

To conclude, this study has some important strengths which may have bearing on future research, such as the identification of the links between mistrust in vaccines, attitudes, perceived behavioural control and intention to vaccinate. These relationships may provide an evidence-based foundation for future research, with the incorporation of monitored psychological factors seemingly becoming increasingly important in vaccine hesitancy research [14]. Further long-term research is also required that evaluates adherence to multiple doses of vaccinations and how younger age groups comply to public health guidance. Finally, the suitability of the Theory of Planned Behaviour in predicting vaccine hesitancy in this study suggests a need for the incorporation of behaviour change theories into public health messaging of vaccines against COVID-19 and future pandemics.

## Author Contributions

**Conceptualization:** Matthew Cavanagh.

**Data curation:** Gavin Breslin, Martin Dempster.

**Formal analysis:** Gavin Breslin, Martin Dempster, Emma Berry, Nicola C. Armstrong.

**Methodology:** Martin Dempster.

**Project administration:** Gavin Breslin.

**Resources:** Martin Dempster, Nicola C. Armstrong.

**Software:** Martin Dempster.

**Supervision:** Gavin Breslin, Emma Berry, Nicola C. Armstrong.

**Validation:** Emma Berry.

**Writing – original draft:** Gavin Breslin, Matthew Cavanagh.

**Writing – review & editing:** Gavin Breslin, Emma Berry, Matthew Cavanagh, Nicola C. Armstrong.

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
