## [Decision Letter · Decision Letter 0]

19 Jul 2021

PONE-D-21-16573

COVID-19 Vaccine Uptake and Hesitancy Survey in Northern Ireland and Republic of Ireland:  Applying the Theory of Planned Behaviour

PLOS ONE

Dear Dr. Cavanagh,

Thank you for submitting your manuscript to PLOS ONE. After careful consideration, we feel that it has merit but does not fully meet PLOS ONE’s publication criteria as it currently stands. Therefore, we invite you to submit a revised version of the manuscript that addresses the points raised during the review process.

We look forward to receiving your revised manuscript.

Kind regards,

Prof. Anat Gesser-Edelsburg, Ph.D.

Academic Editor

PLOS ONE

Journal Requirements:

2. Thank you for including your ethics statement: 'Ethical approval was granted by Ulster University. All participants provided informed consent and were free to withdraw at any time. No personal identifying data was collected to ensure confidentiality.'

Reviewers' comments:

Reviewer's Responses to Questions

**Comments to the Author**

1. Is the manuscript technically sound, and do the data support the conclusions?

Reviewer #1: Partly

Reviewer #2: Yes

2. Has the statistical analysis been performed appropriately and rigorously? 

Reviewer #1: Yes

Reviewer #2: Yes

3. Have the authors made all data underlying the findings in their manuscript fully available?

Reviewer #1: Yes

Reviewer #2: Yes

4. Is the manuscript presented in an intelligible fashion and written in standard English?

Reviewer #1: Yes

Reviewer #2: Yes

5. Review Comments to the Author

Reviewer #1: In this manuscript by Cavanagh and colleagues, the authors used the theory of planned behavior to examine COVID-19 vaccine uptake. This is a timely article as many countries are struggling with COVID-19 vaccine uptake during this pandemic and this article has the potential to provide recommendations and suggestions on addressing vaccine hesitancy.

1. The "Aims of the Study" section is a bit confusing because it is a list of aims and isn't clearly connected to the introduction section. It would be helpful if the authors, clearly stated gaps in existing literature and how study aims will fill existing knowledge gaps.

2. The last aim mentions "to consider confidence in giving the COVID-19 vaccine to children..." however, none of the measures focus on vaccine acceptance for children, and there is no mention of this aim in the results or discussion. If this aim is not included in the current paper, then it should be removed from that section.

3. Participant demographics should be in the results section.

4. "Cross-sectional online survey" is not enough information on the research design. Overall, the methods section is missing a lot of information. For example, if social media platforms were used, additional information on the specific platforms used is needed, along with how the study was advertised on these platforms, such as were flyers posted in specific groups? Did the study staff pay social media platforms to advertise the study to specific populations? This section also needs information on incentives provided to participants.

5. Overall, the discussion section seems more like an extensions of the introduction and provides little information about how the results from this study relate to previous studies. Additionally, the authors stated how the aims of this study was to provide suggestions, but suggestions seemed based off of previous literature instead of study results.

Reviewer #2: Preface

The manuscript is well written, and the study well-conducted I very much appreciated the not easy field of why there is reluctance or hesitation to vaccination programs and especially in some age groups also relates to changes in the patient-doctor relationship but also in the communication of vaccination policy by the ministries or political representatives of the different countries in the world.

The Scottish school has previously addressed this issue as well for other vaccinations.

Kennedy C, Gray Brunton C, Hogg R. 'Just that little bit of doubt': Scottish parents', teenage girls’, and health professionals' views of the MMR, H1N1 and HPV vaccines. Int J Behav Med. 2014 Feb;21(1):3-10. DOI: 10.1007/s12529-013-9356-4. PMID: 24198038.

Introduction

It is too rhetorical to put the history of SAR Covid 2 infections and makes the manuscript banal I would immediately introduce the concept of vaccination also citing other examples in which in the last decade in the world and some countries, it is especially in Europe there is a certain reluctance even towards vaccinations already well known as measles, this reluctance is also encountered in booster.

Please see the following manuscripts

Hilton S, Patterson C, Smith E, Bedford H, Hunt K. Teenagers' understandings of and attitudes towards vaccines and vaccine-preventable diseases: a qualitative study. Vaccine. 2013;31(22):2543-2550. doi:10.1016/j.vaccine.2013.04.023

The data collection methodology is corrected, and the results are well depicted.

The statistical analysis was well conducted. Authors should also consider whether to report a multivariate analysis.

In the debate, I would gently stress the fact that we do not yet know the events that the mutation of the virus will determine on its effectiveness and that it is better to explain the truth to those who must undergo vaccination than to be sure of its effectiveness. Too many times we have hesitated in the methods of communication that are not entrusted to the clinicians but politicians and that the patient finds as a reference preferably the family doctor or pediatrician rather than the minister of health.

The authors could add that in the future they will evaluate adherence to vaccination with multiple doses and compliance with other vaccinations that are added to those recommended in the different age groups (guys and elderly and high-risk group)

6. PLOS authors have the option to publish the peer review history of their article (what does this mean?). If published, this will include your full peer review and any attached files.

Reviewer #1: No

Reviewer #2: **Yes: **Paola Di Carlo

---

## [Author Response · Author response to Decision Letter 0]

11 Sep 2021

Journal Requirements: 

Thank you for providing this information, we have updated the article to reflect the style of formatting required. 

2. Thank you for including your ethics statement: 'Ethical approval was granted by Ulster University. All participants provided informed consent and were free to withdraw at any time. No personal identifying data was collected to ensure confidentiality.' 

The full name of the committee has been added. 

This text has been added. 

Thank you for this additional guidance. 

Participants were provided with a description of the aims and rationale for the study and were asked to provide informed consent before taking part in the survey. Participants could not take the survey without consenting. 

This has been completed on Editoroial Manager. Dr Gavin Breslin’s ORCID ID is: 0000-0003-2481-0860. 

Reviewers' comments: 

Reviewer's Responses to Questions 

Comments to the Author 

1. Is the manuscript technically sound, and do the data support the conclusions? 

Reviewer #1: Partly 

Reviewer #2: Yes 

2. Has the statistical analysis been performed appropriately and rigorously? 

Reviewer #1: Yes 

Reviewer #2: Yes 

3. Have the authors made all data underlying the findings in their manuscript fully available? 

Reviewer #1: Yes 

Reviewer #2: Yes 

4. Is the manuscript presented in an intelligible fashion and written in standard English? 

Reviewer #1: Yes 

Reviewer #2: Yes 

5. Review Comments to the Author 

Reviewer #1: In this manuscript by Cavanagh and colleagues, the authors used the theory of planned behavior to examine COVID-19 vaccine uptake. This is a timely article as many countries are struggling with COVID-19 vaccine uptake during this pandemic and this article has the potential to provide recommendations and suggestions on addressing vaccine hesitancy. 

Thank you for the kind comments and to read that the article is timely and of international interest in addressing vaccine hesitancy. 

1. The "Aims of the Study" section is a bit confusing because it is a list of aims and isn't clearly connected to the introduction section. It would be helpful if the authors, clearly stated gaps in existing literature and how study aims will fill existing knowledge gaps. 

The aims of the study section has been adjusted to reflect that there are gaps in the literature. Page 8 line 187-198 

2. The last aim mentions "to consider confidence in giving the COVID-19 vaccine to children..." however, none of the measures focus on vaccine acceptance for children, and there is no mention of this aim in the results or discussion. If this aim is not included in the current paper, then it should be removed from that section. 

The aim has been deleted. 

3. Participant demographics should be in the results section. 

Participant demographics have been moved to the results section, Page 9 281-284. 

4. "Cross-sectional online survey" is not enough information on the research design. Overall, the methods section is missing a lot of information. For example, if social media platforms were used, additional information on the specific platforms used is needed, along with how the study was advertised on these platforms, such as were flyers posted in specific groups? Did the study staff pay social media platforms to advertise the study to specific populations? This section also needs information on incentives provided to participants. 

The Research Design section has been updated with the above recommendations. Page 9 205-208. 

5. Overall, the discussion section seems more like an extensions of the introduction and provides little information about how the results from this study relate to previous studies. Additionally, the authors stated how the aims of this study was to provide suggestions, but suggestions seemed based off of previous literature instead of study results. 

Sentences have been added to the discussion to show how findings informs recommendations. The recommendations using factors from the Theory of Planned behaviour are now articulated in the dicussion section. Page 19 line 356-358, 363-364, 390-391. 

Reviewer #2: Preface 

The manuscript is well written, and the study well-conducted I very much appreciated the not easy field of why there is reluctance or hesitation to vaccination programs and especially in some age groups also relates to changes in the patient-doctor relationship but also in the communication of vaccination policy by the ministries or political representatives of the different countries in the world. 

The Scottish school has previously addressed this issue as well for other vaccinations. 

Kennedy C, Gray Brunton C, Hogg R. 'Just that little bit of doubt': Scottish parents', teenage girls’, and health professionals' views of the MMR, H1N1 and HPV vaccines. Int J Behav Med. 2014 Feb;21(1):3-10. DOI: 10.1007/s12529-013-9356-4. PMID: 24198038. 

Thank you for bringing this article by kennedy et al (2014) to our attention. We have included within the introduction on page 6 and 7 line 140-145. 

Introduction 

It is too rhetorical to put the history of SAR Covid 2 infections and makes the manuscript banal I would immediately introduce the concept of vaccination also citing other examples in which in the last decade in the world and some countries, it is especially in Europe there is a certain reluctance even towards vaccinations already well known as measles, this reluctance is also encountered in booster. 

Please see the following manuscripts 

Hilton S, Patterson C, Smith E, Bedford H, Hunt K. Teenagers' understandings of and attitudes towards vaccines and vaccine-preventable diseases: a qualitative study. Vaccine. 2013;31(22):2543-2550. doi:10.1016/j.vaccine.2013.04.023 

The study by Hilton has been incorporated into the introduction of the article on page 6. 

The data collection methodology is corrected, and the results are well depicted. 

Thank you for your positive comment. 

The statistical analysis was well conducted. Authors should also consider whether to report a multivariate analysis. 

A multiple linear regression analysis was calculated to incorporate the predictor of vaccine intention Page 17 line 336. 

In the debate, I would gently stress the fact that we do not yet know the events that the mutation of the virus will determine on its effectiveness and that it is better to explain the truth to those who must undergo vaccination than to be sure of its effectiveness. Too many times we have hesitated in the methods of communication that are not entrusted to the clinicians but politicians and that the patient finds as a reference preferably the family doctor or pediatrician rather than the minister of health. 

We have incorporated these points in the discussion, page 21 line 403-405, also see page 21 407-411 

The authors could add that in the future they will evaluate adherence to vaccination with multiple doses and compliance with other vaccinations that are added to those recommended in the different age groups (guys and elderly and high-risk group) 

This has been included page 23 line 456-458 

6. PLOS authors have the option to publish the peer review history of their article (what does this mean?). If published, this will include your full peer review and any attached files. 

Do you want your identity to be public for this peer review? For information about this choice, including consent withdrawal, please see our Privacy Policy. 

Reviewer #1: No 

Reviewer #2: Yes: Paola Di Carlo

---

## [Decision Letter · Decision Letter 1]

19 Oct 2021

COVID-19 Vaccine Uptake and Hesitancy Survey in Northern Ireland and Republic of Ireland:  Applying the Theory of Planned Behaviour

PONE-D-21-16573R1

Dear Dr. Cavanagh,

We’re pleased to inform you that your manuscript has been judged scientifically suitable for publication and will be formally accepted for publication once it meets all outstanding technical requirements.

Kind regards,

Prof. Anat Gesser-Edelsburg, Ph.D.

Academic Editor

PLOS ONE

Additional Editor Comments (optional):

Reviewers' comments:

Reviewer's Responses to Questions

**Comments to the Author**

1. If the authors have adequately addressed your comments raised in a previous round of review and you feel that this manuscript is now acceptable for publication, you may indicate that here to bypass the “Comments to the Author” section, enter your conflict of interest statement in the “Confidential to Editor” section, and submit your "Accept" recommendation.

Reviewer #1: All comments have been addressed

Reviewer #2: All comments have been addressed

2. Is the manuscript technically sound, and do the data support the conclusions?

Reviewer #1: Yes

Reviewer #2: Yes

3. Has the statistical analysis been performed appropriately and rigorously? 

Reviewer #1: Yes

Reviewer #2: Yes

4. Have the authors made all data underlying the findings in their manuscript fully available?

Reviewer #1: Yes

Reviewer #2: Yes

5. Is the manuscript presented in an intelligible fashion and written in standard English?

Reviewer #1: Yes

Reviewer #2: Yes

6. Review Comments to the Author

Reviewer #1: The authors addressed all previous reviewer comments. The deletion of the last aim helps to clarify the objectives of the article. The additional information in the research design section was also helpful to better understand how participants were recruited for the study.

Reviewer #2: I have suggested only minor revisions, and the authors have sufficiently replied to my comments............

7. PLOS authors have the option to publish the peer review history of their article (what does this mean?). If published, this will include your full peer review and any attached files.

Reviewer #1: No

Reviewer #2: **Yes: **Paola Di Carlo

---

## [Editor Report · Acceptance letter]

8 Nov 2021

PONE-D-21-16573R1 

COVID-19 Vaccine Uptake and Hesitancy Survey in Northern Ireland and Republic of Ireland:  Applying the Theory of Planned Behaviour 

Dear Dr. Breslin:

I'm pleased to inform you that your manuscript has been deemed suitable for publication in PLOS ONE. Congratulations! Your manuscript is now with our production department. 

Kind regards, 

on behalf of

Prof. Anat Gesser-Edelsburg 

Academic Editor

PLOS ONE